# Long-Term Outcomes after Non-Traumatic Out-of-Hospital Cardiac Arrest in Pediatric Patients: A Systematic Review

**DOI:** 10.3390/jcm11175003

**Published:** 2022-08-26

**Authors:** Zi Hui Celeste Ng, Siyuan Joseph Ho, Tan Caleb, Clyve Yu Leon Yaow, Seth En Teoh, Lai Peng Tham, Marcus Eng Hock Ong, Shu-Ling Chong, Andrew Fu Wah Ho

**Affiliations:** 1Yong Loo Lin School of Medicine, National University of Singapore, Singapore 117597, Singapore; 2Children’s Emergency, KK Women’s and Children’s Hospital, Singapore 229899, Singapore; 3Department of Emergency Medicine, Singapore General Hospital, Singapore 169608, Singapore; 4Pre-Hospital and Emergency Research Centre, Health Services and Systems Research, Duke-NUS Medical School, Singapore 169857, Singapore; 5Duke-NUS Medical School, Singapore 169857, Singapore

**Keywords:** out-of-hospital cardiac arrest, OHCA, sudden cardiac arrest, survival, mortality, quality of life, pediatrics

## Abstract

Long-term outcomes after non-traumatic pediatric out-of-hospital cardiac arrest (OHCA) are not well understood. This systematic review aimed to summarize long-term outcomes (1 year and beyond), including overall survival, survival with favorable neurological outcomes, and health-related quality of life (HRQoL) outcomes) amongst pediatric OHCA patients who survived to discharge. Embase, Medline, and The Cochrane Library were searched from inception to October 6, 2021. Studies were included if they reported outcomes at 1 year or beyond after pediatric OHCA. Data abstraction and quality assessment was conducted by three authors independently. Qualitative outcomes were reported systematically. Seven studies were included, and amongst patients that survived to hospital discharge or to 30 days, longer-term survival was at least 95% at 24 months of follow up. A highly variable proportion (range 10–71%) of patients had favorable neurological outcomes at 24 months of follow up. With regard to health-related quality of life outcomes, at a time point distal to 1 year, at least 60% of pediatric non-traumatic OHCA patients were reported to have good outcomes. Our study found that at least 95% of pediatric OHCA patients, who survived to discharge, survived to a time point distal to 1 year. There is a general paucity of data surrounding the pediatric OHCA population.

## 1. Introduction

Out-of-hospital cardiac arrest (OHCA) is the most time-critical medical emergency and exerts a tremendous disease burden [1,2]. With increasingly sophisticated clinical and systems interventions designed to optimize the care of OHCA, more patients who experience OHCAs are surviving the initial event. There is tremendous scientific and public health interest in the long-term survivorship of OHCA patients, with unanswered questions on the duration and quality of survival, and their determinants [3,4].

Recent attempts to synthesize the available literature on long-term outcomes after OHCA have found that amongst patients who survived to hospital discharge or to 30 days, about half survived to 5 years [5]. They have also found a large burden of psychological comorbidities developing amongst the survivors [6]. Regrettably, these systematic reviews have excluded studies on pediatric patients in order to reduce issues with clinical heterogeneity. Moreover, the same instruments used to characterize health-related quality of life (HRQOL) in adults are rarely applicable to children. In contrast to adult OHCA, pediatric OHCA is typically unwitnessed and precipitated by events of non-cardiac etiologies, such as respiratory arrest, sudden infant death syndrome, and drowning, and present with initial asystole [7,8,9].

Yet, unique to the pediatric population is a general impression that their young age is associated with a greater brain plasticity [10] that makes the brain better able to withstand and recover from an acute insult. This is attributable to the “Kennard Principle” [11], which details predictive factors for functional outcomes, including having a brain lesion at a younger age being associated with better functional outcomes. In fact, recent literature suggests that survival to hospital discharge rates in pediatric OHCA patients of 8.2% [12] to 8.6% [9] are comparable to the survival to hospital discharge rates of 8.8% [13] in adult OHCA patients.

Therefore, knowledge of the long-term outcomes after pediatric OHCA is keenly needed but currently poor [14]. In the Pediatric Utstein Style template [15], a standardized reporting framework for resuscitation registries—survival to discharge was a core outcome, while survival at 1 year was a supplemental outcome. Moreover, Topjian et al. [14] also detail the many gaps in knowledge surrounding the topic of pediatric OHCA, including best practices to monitor neurobehavioral and quality-of-life measures amongst survivors.

Hence, we aimed to evaluate the long-term outcomes (1 year and beyond) including overall survival and survival with favorable neurological outcomes, as well as the health-related quality-of-life (HRQoL) outcomes, amongst pediatric OHCA patients who survived to discharge.

## 2. Materials and Methods

This systematic review and meta-analysis adhered to the Preferred Reporting Items for Systematic Reviews and Meta-Analyses (PRISMA) guidelines [16]. The study protocol has been registered in the International Prospective Register of Systematic Reviews.

### 2.1. Search Strategy

A systematic literature search was performed in Medline, Embase, and Cochrane CENTRAL databases from inception up to 6 October 2021. The search strategy was developed in consultation with a medical information specialist (Medical Library, National University of Singapore). Keywords and MeSH terms such as “Out-of-hospital Cardiac Arrest”, “Survival”, “Mortality”, “Quality of Life”, and “Pediatrics” were used in the search strategy. The detailed search strategy is available in the Appendix A). Content experts were consulted for additional references and references of relevant sources were hand-searched to identify additional relevant studies. Articles were viewed on Endnote X9 (Clarivate, Philadelphia, PA, USA) for the sieving of articles.

### 2.2. Inclusion and Exclusion Criteria

Article sieve was conducted by three authors (ZHCN, SJH, TC) according to predefined criteria. Each article was reviewed by at least two authors blinded to each other’s decision. Disputes were resolved through consensus from the senior author (AFWH). The predefined criterion for inclusion were: (1) studies with pediatric patients (age < 21 years) with OHCA of nontraumatic etiology, (2) articles reporting outcomes of pediatric OHCA patients at 1 year and beyond, (3) study design was cohort or cross-sectional, and (4) articles written or translated into the English language. The criterion for exclusion were: (1) studies where data for OHCA patients could not be isolated from in-hospital cardiac arrest patients, (2) studies where data for pediatric patients could not be isolated from adult patients (age ≥ 21 years), (3) articles without primary data (reviews, systematic reviews, commentaries, and editorials), and (4) studies examining OHCA in a highly selective subgroup of patients (i.e., liver failure and pregnant patients). To prevent duplication of patient datasets analyzed, studies originating from the same center(s) during the same or overlapping time periods were collated, and only the most relevant study was included for analysis.

### 2.3. Data Abstraction

Data were abstracted into an Excel spreadsheet (Microsoft Corp, New Mexico, United States). Each article was triple-coded by three authors (ZHCN, SJH, TC). Disputes were resolved through consensus from the senior author (AFWH). Data abstracted included study characteristics (e.g., author name, year of publication, country, region), study population characteristics (e.g., sample size, ethnicity, age, country, comorbidities), emergency medical services (EMS) system characteristics (e.g., single or dual dispatch system), and OHCA event characteristics (e.g., location, initial rhythm, etiology, use of targeted temperature management (TTM), cause of death, survival). Shockable rhythms were defined as ventricular fibrillation and pulseless ventricular tachycardia. Etiologies were classified into medical, drug overdose, drowning, electrocution, and asphyxia. Whenever relevant, data abstraction was organized according to the Utstein style for reporting OHCA cases [17]. Survival outcomes included survival and survival with favorable neurological status at 1 year and in the longer term (a time point distal to 1 year). Favorable neurological status was defined with the Pediatric Outcome Performance Category/ Pediatric Cerebral Performance Category [18] which is stratified into 6 categories, with scores of 1–6 corresponding to the various categories: normal (age-appropriate functioning), mild disability, moderate disability, severe disability, coma or vegetative state, and finally, death. We considered a score of either 1 or 2 as fulfilling favorable neurological status.

For continuous variables, mean and standard deviation (SD) were abstracted. Where these data were unavailable, appropriate formulae were applied to transform the data from median and range or interquartile range to mean and SD [19,20]. For categorical variables, frequency and percentages were abstracted.

### 2.4. Statistical Analysis

Where deemed appropriate, we had planned to utilize a single-arm meta-analysis of proportions to summarize survival and survival with favorable neurological outcomes at discharge, at 1 year and in the longer term (a time point distal to 1 year). For data that had fewer than three data points, meta-analysis was considered to be inappropriate. After assessing extracted data, a meta-analysis was deemed inappropriate due to too much heterogeneity and insufficient data points. These were therefore described in the systematic review.

### 2.5. Risk of Bias Assessment

The risk of bias assessment was conducted using the Newcastle–Ottawa Scale (NOS) [21]. The NOS assesses each study in three domains: selection of the research population, comparability of the study groups, and results. NOS scores ≥ 7 stars were considered high-quality.

## 3. Results

### 3.1. Literature Retrieval and Summary of Included Articles

The database search yielded 722 articles, whilst the search of reference lists of references yielded 33 articles. One hundred two duplicated articles were removed. Five hundred eighty articles were then excluded based on their titles and abstracts. A further 67 articles were excluded upon full-text review. One paper was included upon content expert consultation. Finally, seven studies qualified for analysis. The study selection process and reasons for excluding studies are illustrated in the PRISMA-P 2020 flow diagram (Figure 1).

A total of 2842 patients were included across these seven studies. Mean age ranged from 1.5 to 6.4 years. Three studies were conducted in the United States of America [22,23,24], two were conducted in the Netherlands [25,26], one in Finland [27], and one in Taiwan [28]. Two studies were prospective cohort studies, and five were retrospective cohort studies. Follow-up periods ranged from 2 to 8.5 years. All studies were classified to be of moderate to high quality. The characteristics and quality assessment of the included studies are available in Appendix A. Six out of seven studies provided data on survival to discharge, and this ranged from 3 to 89%. Four out of seven studies provided data on survival to hospital discharge with favorable neurological outcomes, and this ranged from 2 to 62%.

### 3.2. Survival Outcomes beyond One Year

In terms of survival beyond 1 year, only two [25,26] studies reported this outcome. Albrecht et al. reported a survival of 95% at a median follow-up time of 26.3 months, while Hunfeld et al. reported a survival of 98% at a mean follow-up time of 24 months.

In terms of survival with favorable neurological outcomes beyond 1 year, only two [25,26] studies reported this outcome. Albrecht et al. reported that 10% of patients had a favorable neurological outcome at a median follow-up time of 3.7 years, while Hunfeld et al. reported that 71% of patients had a favorable neurological outcome at a mean follow-up time of 2 years. Both authors utilized the PCPC for reporting these outcomes.

Only one [25] study reported outcomes for patients stratified by age. Albrecht et al. reported stratified analyses at several time points, with this paper taking the time point at the longest follow-up into consideration: 27% of infants (aged < 1 year), 32% of children (aged 1–11 years), and 38% of adolescents (aged 12–18 years) were reported to have favorable neurological outcome beyond one year.

The summary of studies for each of the above-mentioned outcomes is presented in Table 1.

### 3.3. Health-Related Quality of Life Outcomes

Four articles reported the HRQoL outcomes of pediatric OHCA patients. One [26] study was comparative, comparing the HRQoL of the long-term survivors of pediatric OHCA to the general population, while the other three [23,24,27] studies were non-comparative. The summary of the four studies is presented in Table 2.

Hickson et al. [24] utilized the PedsQL [29] to measure HRQoL outcomes of pediatric OHCA patients on a scale of 0–100, with a higher score indicating better HRQoL. PedsQL scores for the studied population were reported as 85 (95% CI = 69, 93). PedsQL scores were reported to be lower for children with caregiver-described limitations (PedsQL = 71, PedsQL–FIM = 64 [30]) than children without limitations (PedsQL = 92, PedsQL–FIM = 95).

#### 3.3.1. Neurocognitive and Neuropsychological Outcomes

Hunfeld et al. [26] utilized 11 instruments: Total IQ, Verbal IQ, Performance IQ, Selective Attention (STROOP ≥ 11 years), Sustained Attention (Bourdon SD ≥ 6 years), Processing Speed (≥4 years), VMI (Beery ≥ 2 years), Verbal Memory (Rey-AVLT, delayed recall ≥ 6 years), Visual Memory (ReyRecog ≥ 5 years), Cognitive Flexibility (TMTB ≥ 8 years), and BRIEF Total score (≥2 years). All neuropsychological tests were converted into Z-scores and outcomes were elicited at 3–6 months and 24 months, with good outcomes (PCPC 1–2) reported in 74 and 73% of survivors, respectively. It was further reported that there were no significant changes in neuropsychological outcomes found in pediatric OHCA patients from the first time point to the next.

Suominen et al. [27] first utilized eight instruments: four verbal and four performance Wechsler subtests, and then extrapolated these intelligence scales to derive the verbal and performance IQ and full-scale IQ (FIQ). Sixty percent of the survivors had a FIQ of more than 80, which was taken to be the cutoff for what constitutes low IQ. Additionally, there were no significant differences between the mean performance and verbal IQ scores in all the subjects (*p* = 0.85).

Hunfeld et al. [26] and Suominen et al. [27] both found that intelligence scores of pediatric OHCA did not correlate with their PCPC scores. Specific to Total IQ, Hunfeld et al. [26] found that long-term survivors of pediatric OHCA did not have comparable outcomes as compared to the general population, with a Z-score of −0.1 and −0.3 at 3–6 months and 24 months, respectively. Suominen et al. [27], however, did not have a standardized measure but found that 40% of long-term survivors of pediatric OHCA had low IQ.

#### 3.3.2. Functional Outcomes

Silka et al. [23] utilized the Neurological Impairment Scale [31], which is meant to predict functional outcomes in neurorehabilitation. During 72 ± 37 months of follow-up, 38% of survivors had a normal neurologic outcome, while 32 and 30% of survivors had a mild and moderate-to-severe neurologic impairment, respectively. Taking normal and mild neurologic impairment as good outcomes, 70% of survivors have had a good outcome.

### 3.4. Post-discharge Diagnosis

Only one paper reported outcomes on post-discharge diagnosis (PDD) at 1 year post-discharge. Lee et al. [28] reported that the five most common PDD types were: respiratory tract diseases (72.2%), gastrointestinal diseases (50.0%), neurological disease (49.1%), skin or soft tissue diseases (42.5%), and eye/ ear diseases (26.5%). Further, pneumonia (22.7%), acute gastroenteritis (34.8%), and epilepsy (20.6) respectively constituted the majority of respiratory, gastrointestinal, and neurological diseases.

### 3.5. Family and Caregiver Burden

Only one paper reported outcomes on family and caregiver burden in the longer term. Hickson et al. [24] employed the PedsQL family impact module (PedsQL–FIM [30]), a specific module of the PedsQL, in measuring the impact of pediatric acute and chronic health conditions on parents and the family (i.e., caregivers). PedsQL–FIM scores were 76 (95% CI = 63, 95).

## 4. Discussion

This systematic review describes the long-term quantitative and qualitative outcomes of pediatric non-traumatic OHCA patients. To our knowledge, this is the first systematic review addressing this question. The main findings, with the caveat of limited datasets, were as follows: longer-term survival was at least 95%, while survival with favorable neurological outcomes beyond a year post-OHCA was heterogeneous across studies, ranging from 10 to 71%. With regard to health-related quality of life outcomes, at a time point distal to 1 year, at least 60% of pediatric non-traumatic OHCA patients were reported to have good outcomes.

This analysis found that the survival in the longer term was at least 95%. This high proportion shows that long-term outcomes are better in children compared to adults. However, in both pediatric and adult studies, there exists a possibility that these outcomes are inflated by a selection bias, in that high-performing health systems are more likely to report their findings. This may be due to having a cardiac arrest data collection workflow (which is a feature of a developed emergency medical system). The studies of Albrecht et al. [25] and Hunfeld et al. [26] were both conducted in the Netherlands, and the protocols taken by the Dutch Emergency Medical Services (EMS) have been detailed in considerable thoroughness by Bardai et al. [32]. From individuals dialing the national emergency number to EMS response with personnel equipped with manual defibrillators and qualified in performing advanced life support, the Netherlands has demonstrated that it has a well-developed EMS. This sentiment is echoed by de Visser et al. [33], whose paper suggests that the optimization of the chain of survival, of which EMS response is a link, may be of benefit to the survival of OHCA patients. This finding, however, might not be generalizable to countries with poorer EMS responses [34,35], regional variation [36], or poorer socioeconomic composition [37].

The data available did not permit a meta-analysis of long-term survival with favorable neurological outcomes. Albrecht et al. [25] found that only 10% of patients would have favorable neurological outcomes in the longer term, whilst Hunfeld et al. [26] reported a rosier 71%. Albrecht et al. [25] reported numerous timings and sources of when the long-term neurological outcome was determined, with this analysis considering only that of the longest follow-up time. At each scoring time point, the number of patients assessed was variable and was not equivalent to the total number of patients discharged, which could have caused the data to be unreflective of the true percentage. For this reason, the disparity between the two studies resulted in an inconclusive finding. To triangulate the data, Michels et al. [38] had 50% of patients having favorable neurological outcomes in the long-term follow-up. It is hence likely that longer-term survival with favorable neurological outcomes is likely to have fairly optimistic results, but more research is necessary to confirm this postulation.

Additionally, our analysis found that infants <1 year and children 0–5 years had the lowest percentage of patients who survived with favorable neurological outcomes both at discharge and in the long term. Indeed, this is echoed by Kitamura et al. [39], who reported that survival with favorable neurological outcomes in infants was uniformly poor, at 1.7%. This is further supported by Atkins et al. [8], who reported a mean overall survival of 3.4% in infants, as opposed to that of 8.5 and 8.2% in children and adolescents, respectively. These statistics may be supported by the plasticity of the growing brain, whereby there is global brain damage in infants from hypoxia and hypotension, as opposed to older children and adults. Moreover, there is differential training for and exposure to infant resuscitation in EMS systems, which might limit the effectiveness of the first responders. This is supported by Atkins et al. [8], who showed that interventions by EMS were not as effective for infants as they were in other age groups, as the number needed to treat to save a life was 29 for infants, as opposed to the 13 in the general pediatric population. Topjian et al. [14] further identify critical knowledge gaps with regard to age-dependent mechanisms as a consideration.

It is worth noting that for adult OHCA patients who had survived to 1 year, 83.3% [5] of patients had favorable neurological outcomes. The numbers in the pediatric population pale in comparison to this. This could be attributed to a lack of interventions proven to work for the pediatric OHCA population. Moler et al. [40] published results of a large-center trial which showed no difference between the neurological outcomes at 1 year for children who had and who had not been subjected to therapeutic hypothermia. Similarly, while Wolf et al. [41] have demonstrated the potential of using extracorporeal cardiopulmonary resuscitation in improving outcomes for in-hospital pediatric cardiac arrest, this has not yet been evaluated for the pediatric OHCA population. Finally, Topjian et al. [14] also detail the many gaps in knowledge surrounding the topic of pediatric OHCA, including the role of implementing a bundle of care, the role of pediatric cardiac arrest centers in optimizing outcomes, and the best measures for monitoring neurobehavioral and quality-of-life measures in the patient. This is in stark contrast to the ample literature on adult OHCA [42,43] which has answered these questions.

Considering the gravity of OHCA, sequential diseases and care of the diseases should be accounted for. Lee et al. [28] reported that most of the post-discharge diseases, including almost half of the neurological, urological, and cardiovascular events, were newly diagnosed within the first 3 months after discharge. Subsequently, dermatitis, the main skin problem for survivors, was more commonly diagnosed 4–6 months after discharge. This is possibly attributable to the onset of a transient dysfunction between the OHCA event and discharge. Indeed, Murkin et al. [44] reported that the incidence of postoperative neurobehavioral dysfunction is highest in the immediate postoperative period, before declining. Considering the thin amount of data on post-discharge diseases, more research should be done to determine the pathways of such transient dysfunctions.

While Hickson et al. [24] reported a PedsQL–FIM score of 76 (a scale of 0–100, with higher scores indicating a better HRQoL), Hickson et al. utilized only a small pilot sample size of 17, which Bohm et al. [45] cautioned against. In addition, both Meert et al. [46] and Bohm et al. [45] found that poor neurological outcomes are related to increased caregiver burden. With the postulation that longer-term survival with favorable neurological outcomes is likely to have fairly optimistic results, caregiver burden may not be severe for the caregivers of the pediatric OHCA population. Considering the thin amount of data on caregiver burden, more research should be done to explore its relation to the pediatric OHCA population. Support groups can also be recommended to these caregivers.

### Strengths and Limitations

Our findings should be interpreted in the context of the following strengths: Firstly, to our best knowledge, this is the most extensive systematic review summarizing the long-term outcomes of pediatric OHCA patients. Secondly, this paper has scoped the study specifically to that of the non-traumatic pediatric OHCA population, with no overlaps with any of the converse populations—traumatic, adult, or in-hospital cardiac arrest populations.

However, this systematic review concedes some limitations. Owing to the lack of representation from the Asian, African, South American, and Middle Eastern regions, our estimates are unlikely to reflect global long-term outcomes of pediatric OHCA cases. Next, only articles written in or translated into the English language were included in this paper, which further limits the generalizability of the paper. Third, there was a general paucity of long-term outcomes for pediatric OHCA patients. Lastly, our results may not accurately reflect the long-term outcomes of patients with non-shockable rhythms, as Albrecht et al. [25] included only patients with shockable rhythm.

While randomized controlled trials were excluded from this meta-analysis, we noted in our sieve that a large proportion of papers pertaining to pediatric OHCA were based on a multi-center study, Therapeutic Hypothermia after Pediatric Cardiac Arrest (THAPCA) [40], conducted in 2015. This would have greatly skewed the analysis of the data had we included it, thereby further validating the need for greater research in this field.

This systematic review found that beyond the paucity of data, there is also no consensus on age cutoffs. There is hence a need to identify and decide on appropriate age cutoffs (e.g., infants are <1 year old, children are 1–11 years old, adolescents are 12–18 years old, with the cutoff for the pediatric population being below 19 years old). Such information would make subgroup analysis more robust and enable a more comprehensive analysis of long-term outcomes in pediatric OHCA patients in the future.

We found that a variety of scales were used to assess the patients’ neurological outcomes, which limits the ability of clinicians to compare the different studies. Unlike the widely used Cerebral Performance Categories (CPC) [43] in assessing neurological outcomes in adults, we found that the pediatric population had a more diverse pool of measures. The Pediatric Outcome Performance Categories (POPC) and Pediatric Cerebral Performance Categories (PCPC) were considered congruent [18] and similar to the Functional Status Scale [47], but the Pediatric Index Mortality 3 [48] has not been determined to serve as an equivalent. There is hence a need to identify and develop a singular standardized scale for reporting and comparing the neurological outcomes in pediatric OHCA patients to allow for better comparisons across different studies and populations. This would enable a more uniform measure and analysis of neurological outcomes in pediatric OHCA patients in the future.

Additionally, we found that a variety of scales and instruments were utilized to assess the patients’ HRQoL, which limits the ability of clinicians to compare the different studies. In fact, a recent systematic review by Haywood et al. [49] concluded a lack of HRQoL research in the OHCA population. On top of the dismal number of papers which results in little being known about the long-term neuropsychological function in OHCA survivors, as reported by van Zellem et al. [50], the lack of comparative validated scores led to a difficulty in interpretation. Moreover, the papers did not report the numerical values of the HRQoL scores, instead having published secondary data which had been analyzed or extrapolated from the raw data, which impeded the possibility of meta-analysis. There is hence a need to identify and develop a standardized method of reporting and comparing the HRQoL of pediatric OHCA patients to allow for better comparisons across different studies and populations. This would enable a more uniform measure and analysis of HRQoL data in pediatric OHCA patients in the future.

## 5. Conclusions

Our study found that at least 95% of pediatric OHCA patients who survived to discharge survived to a time point distal to 1 year. Further research on the long-term outcomes of pediatric OHCA patients is needed to better direct future breakthroughs in the long-term treatment of pediatric OHCA patients.

## Figures and Tables

**Figure 1 jcm-11-05003-f001:**
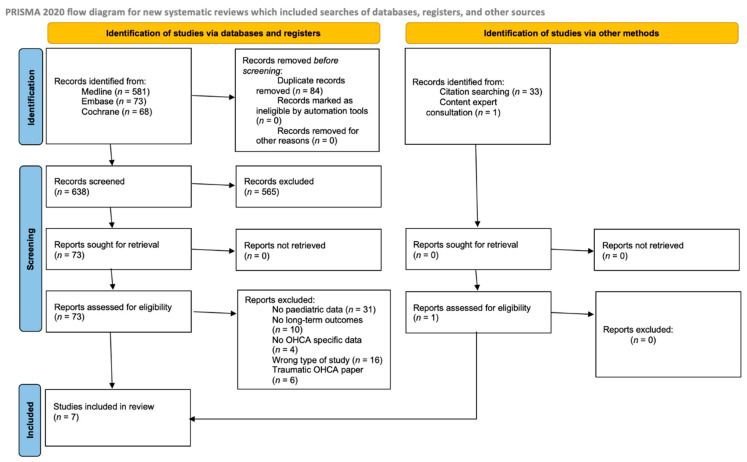
PRISMA-P 2020 flow diagram.

**Table 1 jcm-11-05003-t001:** Survival outcomes beyond one year.

Survival Outcomes beyond One Year					
**Paper**	**Follow-Up Time (Months)**	**Total** **Population**	**Number at** **Follow-Up**	**Events**	**Survived (%)**
Albrecht et al. (2021) [25]	26.3 (median)	142	135	7	95%
Hunfeld et al. (2021) [26]	24	49	48	1	98%
**Survival Outcomes beyond One Year with Favorable Neurological Outcomes**					
**Paper**	**Follow-Up Time (Months)**	**Total** **Population**	**Number at** **Follow-Up**	**Number with Favorable** **Neurological Outcome (%)**	
Albrecht et al. (2021) [25]	44.4 (median)	142	14	10	
Hunfeld et al. (2021) [26]	2	49	35	71	
**Survival Outcomes beyond One Year with Favorable Neurological Outcomes, by Age**					
**Paper**	**Follow-Up Time (Months)**	**Age Group**	**Total Population**	**Number at Follow-Up**	**Number with Favorable Neurological Outcome (%)**
Albrecht et al. (2021) [25]	28.3 (median)	Infants (<1 year)	95	26	27
	23.1 (median)	Children (1–11 years)	187	60	32
	25.7 (median)	Adolescents (12–18 years)	77	29	38

**Table 2 jcm-11-05003-t002:** Health-Related Quality of Life Outcomes.

Author	Year	Scales Used	Further Explanation	Summary Estimates
Hickson et al. [24]	2021	PedsQL	85 (69, 93)	PedsQL scores were 85 (69, 93) and did not differ from normative controls (*p* = 0.09)
Hunfeld et al. [26]	2021	Intellectual Functioning: Total IQ Age-appropriate versions of the Bayley Scales of Infant Development or the Wechsler Scales (BSID-cognitive score, WPPSI-III TIQ, WISC-III TIQ, or WAIS-IV TIQ)	−0.4 (−1.5 to 0.2) *p* vs. norm = 0.008	*p* vs. norm = 0.008 At 24 months, compared to the norm, OHCA survivors had worse scores for intellectual functioning compared with norm data Lower intelligence scores were found at 24 months in critically ill PICU survivors No significant changes to intelligence scores and neuropsychological outcome were detected over time with repeated assessment
		Intellectual Functioning: Verbal IQ Age-appropriate versions of the Bayley Scales of Infant Development or the Wechsler Scales (WPPSI-III VIQ, WISC-III VIQ, WAIS-IV VC-index)	−0.5 (−1.6 to 0.4)	*p* vs. norm = 0.05 At 24 months, compared to the norm, OHCA survivors had worse scores for intellectual functioning compared with norm data Lower intelligence scores were found at 24 months in critically ill PICU survivors No significant changes to intelligence scores and neuropsychological outcome were detected over time with repeated assessment
		Intellectual Functioning: Performance IQ Age-appropriate versions of the Bayley Scales of Infant Development or the Wechsler Scales (WPPSI-III PIQ, WISC-III PIQ, WAIS-IV PO-index)	−0.5 (−2.0 to 0.0)	*p* vs. norm = 0.02 At 24 months, compared to the norm, OHCA survivors had worse scores for intellectual functioning compared with norm data Lower intelligence scores were found at 24 months in critically ill PICU survivors No significant changes to intelligence scores and neuropsychological outcome were detected over time with repeated assessment
		Selective Attention (STROOP ≥ 11 y) Stroop Color Word Test	−1.3 (−1.6 to −0.5)	*p* vs. norm = 0.02 At 24 months, compared to the norm, OHCA survivors had worse scores for selective attention
		Sustained Attention (Bourdon SD ≥ 6 y) Bourdon Vos Cancellation Test	−4.7 (−7.4 to −2.2)	*p* vs. norm = 0.002
		Processing Speed (≥ 4 y) Wechsler Scales (WPPSI-III, WISC-III or WAIS-IV)	−1.0 (−1.8 to 0.0)	*p* vs. norm = 0.003 At 24 months, compared to the norm, OHCA survivors had worse scores for processing speeds compared with norm data
		Visual Motor Integration (Beery ≥ 2 y) Beery Developmental Test of Visual Motor Integration	−0.7 (−1.1 to 0.2)	*p* vs. norm = 0.08
		Verbal Memory (Rey-AVLT, delayed recall ≥ 6 y) Rey Auditory Verbal Learning Test, Delayed Recall	0.2 (−1.1 to 1.1)	*p* vs. norm = 0.82
		Visual Memory (ReyRecog ≥ 5 y) Rey–Osterrieth Complex Figure Test Recognition	–0.4 (−0.8 to 0.2)	*p* vs. norm = 0.28
		Cognitive Flexibility (TMTB ≥ 8 y) Trail-Making Test part B	−1.2 (−2.0 to −0.1)	*p* vs. norm = 0.04 At 24 months, compared to the norm, OHCA survivors had worse scores for cognitive flexibility functioning compared with norm data
		BRIEF Total score (≥ 2 y) Behavior Rating Inventory of Executive Function Questionnaires (BRIEF-P or BRIEF)	0.0 (−1.0 to 0.4)	*p* vs. norm = 0.41
Silka et al. [23]	2018	Neurological Impairment Scale	No residual neurological impairment compared to pre-arrest status (*n* = 15) Mild levels of impairment in either high level cognitive or physical function (*n* = 13) Moderate levels of impairment (*n* = 6) Severe levels of impairment (*n* = 6)	Patients with VF had a higher proportion of good neurological outcomes compared to patients with asystole (17 vs. 2%)
Suominen et al. [27]	2014	Wechsler Intelligence Manual Scales to assess for higher cortical function such as IQ WPPSI-III (Ages 3–7) WISC-III (Ages 7–16) WAIS-III (Ages > 16) NEPSY-II for ages 3–16 to analyze neurocognitive function	Full-scale IQ (FIQ) < 80 (*n* = 8) ○ 4 had FIQ < 70 (intellectual disability) FIQ > 80 (*n* = 12) ○ 5 patients had neurological deficit in memory, executive function, or both	Patients who received CPR from EMS units had a higher risk of major neurological dysfunction (*p* = 0.006) and low FIQ (*p* = 0.017) compared to patients who only required bystander CPR Submersion time, base excess on arrival to ER, mechanical ventilation time and length of stay in the PICU were also associated with neurological and cognitive outcomes Submersion time (mins) ○ Neurologically intact (4.0 (2.0, 6.3)) vs. minor or major neurological deficit (7.5 (4.0, 19.0)), *p* = 0.037 ○ Normal FIQ (3.5 (2.0, 7.5)) vs. low FIQ (12.5 (5.0, 22.5)), *p* = 0.013 Base excess (mmol/L) ○ Neurologically intact (−7.6 (−15.3, −4.0)) vs. minor or major neurological deficit (−17.0 (−23.2, −10.6)), *p* = 0.047 ○ Normal FIQ (−8.7 (−16.0, -4.5)) vs. low FIQ (−21.0 (−25.4, −12.9)), *p* = 0.025 Mechanical ventilation time (days) ○ Neurologically intact (1.0 (1.0, 1.0)) vs. minor or major neurological deficit (3.0 (2.0, 7.0)), *p* = 0.005 ○ Normal FIQ (1.0 (1.0, 2.5)) vs. low FIQ (3.0 (2.5, 6.0)), *p* = 0.051 Length of PICU stay (days) ○ Neurologically intact (1.0 (1.0, 2.0)) vs. minor or major neurological deficit (5.0 (3.0, 7.5)), *p* = 0.005 ○ Normal FIQ (1.0 (1.0, 2.0)) vs. low FIQ (5.5 (4.5, 7.5)), *p* = 0.002

## Data Availability

Not applicable.

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
