# Peer review of "Long-Term Outcomes after Non-Traumatic Out-of-Hospital Cardiac Arrest in Pediatric Patients: A Systematic Review"

_jcm, 2022, doi:10.3390/jcm11175003_

Round 1

Reviewer 1 Report

Thank you for the opportunity to review this paper. The authors have attempted to identify the long-term outcomes of pediatric OHCA patients which is an area that is lacking data overall.

The study is well done. The review highlights key strengths and limitations.

I would suggest editing the conclusion - key to this paper is that data is lacking overall and as you suggest is skewed to North American/Western European systems. I think this needs to be emphasized more in the conclusions, otherwise someone who is skimming through the paper or abstract may well be mislead. Highlighting further by adding a line such as "with the caveat of limited datasets" would be beneficial

Author Response

Dear Reviewer,

Thank you for your review and the comments that you have given. We greatly appreciate it. We have incorporated the feedback (editing the conclusion to emphasise the limited datasets more) into our revised draft. Thank you once again.

Reviewer 2 Report

I appreciate the opportunity to read this interesting manuscript.

I advise reformulating point 2.4 "statistical analysis", since a meta-analysis was not carried out, it should not be described how it was intended to be carried out.

The authors conduct a systematic review of outcomes after non-traumatic out-of-hospital pediatric cardiac arrest.

It is a well-planned and methodologically well-developed work.

Initially, they planned to carry out a meta-analysis, but they finally carried out a systematic review, which they adequately justify.

I advise reformulating point 2.4 "statistical analysis", since a meta-analysis was not carried out, it should not be described how it was intended to be carried out.

I believe that the manuscript is of sufficient quality.

Author Response

Dear Reviewer,

Thank you for your review and the comments that you have given. We greatly appreciate it. We have incorporated the feedback (reformulating point 2.4 "statistical analysis") into our revised draft. Thank you once again.